# Deoxyxylulose 5-Phosphate Synthase Does Not Play a Major Role in Regulating the Methylerythritol 4-Phosphate Pathway in Poplar

**DOI:** 10.3390/ijms25084181

**Published:** 2024-04-10

**Authors:** Diego González-Cabanelas, Erica Perreca, Johann M. Rohwer, Axel Schmidt, Tobias Engl, Bettina Raguschke, Jonathan Gershenzon, Louwrance P. Wright

**Affiliations:** 1Department of Biochemistry, Max Plank Institute for Chemical Ecology, Hans-Knöll-Straße 8, 07745 Jena, Germany; dgc87bueu@gmail.com (D.G.-C.); aschmidt@ice.mpg.de (A.S.); raguschke@ice.mpg.de (B.R.); gershenzon@ice.mpg.de (J.G.); zeiselhof@tutanota.com (L.P.W.); 2Laboratory for Molecular Systems Biology, Department of Biochemistry, Stellenbosch University, Private Bag X1, Matieland, Stellenbosch 7602, South Africa; jr@sun.ac.za; 3Department of Insect Symbiosis, Max Plank Institute for Chemical Ecology, Hans-Knöll-Straße 8, 07745 Jena, Germany; tengl@ice.mpg.de

**Keywords:** isoprene, DXS enzyme, MEP pathway, DMADP, IDP, metabolic control analysis (MCA), flux control coefficient (FCC), isoprenoid

## Abstract

The plastidic 2-C-methylerythritol 4-phosphate (MEP) pathway supplies the precursors of a large variety of essential plant isoprenoids, but its regulation is still not well understood. Using metabolic control analysis (MCA), we examined the first enzyme of this pathway, 1-deoxyxylulose 5-phosphate synthase (DXS), in multiple grey poplar (*Populus* × *canescens*) lines modified in their DXS activity. Single leaves were dynamically labeled with ^13^CO_2_ in an illuminated, climate-controlled gas exchange cuvette coupled to a proton transfer reaction mass spectrometer, and the carbon flux through the MEP pathway was calculated. Carbon was rapidly assimilated into MEP pathway intermediates and labeled both the isoprene released and the IDP+DMADP pool by up to 90%. DXS activity was increased by 25% in lines overexpressing the DXS gene and reduced by 50% in RNA interference lines, while the carbon flux in the MEP pathway was 25–35% greater in overexpressing lines and unchanged in RNA interference lines. Isoprene emission was also not altered in these different genetic backgrounds. By correlating absolute flux to DXS activity under different conditions of light and temperature, the flux control coefficient was found to be low. Among isoprenoid end products, isoprene itself was unchanged in DXS transgenic lines, but the levels of the chlorophylls and most carotenoids measured were 20–30% less in RNA interference lines than in overexpression lines. Our data thus demonstrate that DXS in the isoprene-emitting grey poplar plays only a minor part in controlling flux through the MEP pathway.

## 1. Introduction

Isoprenoids are the largest and structurally most diverse group of low molecular weight metabolites in living organisms, with a multitude of different functions [1,2]. Despite this structural and functional variety, all isoprenoids are derived from the same five-carbon building blocks, isopentenyl diphosphate (IDP) and its isomer dimethylallyl diphosphate (DMADP) [3,4,5,6,7,8]. These five-carbon precursors can be synthesized via two different pathways in plants: the mevalonic acid (MVA) pathway and the 2-C-methyl-methyleritritol-4-phosphate (MEP) pathway [4,5,7,8,9]. Unlike in other living organisms, these two independent pathways coexist within the same cell, but in different compartments. While the cytosolic MVA pathway produces IDP and DMADP as precursors for sesquiterpenes, sterols, brassinosteroids, protein farnesylation, and cytokinin biosynthesis, among others, the plastidic MEP pathway is responsible for the biosynthesis of photosynthesis-related isoprenoids (carotenoids and the side chains of chlorophylls, plastoquinones, and tocopherols), hormones (gibberellins and abscisic acid), and a wealth of monoterpenes and diterpenes [1,10,11,12,13,14]. While previous studies showed a limited exchange of common intermediates between both pathways, this exchange is not capable of rescuing a pharmacological block in either pathway [11,15,16,17], suggesting an important role of the MEP pathway as the source of precursors for essential plastid isoprenoids.

The seven steps of the MEP pathway begin with 1-deoxy-D-xylulose-5-phosphate (DXP) synthase (DXS), which catalyzes the condensation of the precursors glyceraldehyde 3-phosphate (GAP) and pyruvate (PYR) into DXP [18,19]. DXP is then reduced by DXP reductoisomerase (DXR) to 2-C-methyl-D-erythritol-2,4-phospate (MEP) and further converted in three steps to 2-C-methyl-D-erythritol-2,4-cyclodiphosphate (MEcDP). MEcDP is successively reduced to (*E*)-4-hydroxy-3-methylbut-2-enyl 4-diphosphate (HMBDP), catalyzed by HMBDP synthase (HDS), and then to a mixture of IDP and DMADP, catalyzed by HMBDP reductase (HDR) [20,21] (Figure 1). Some of these intermediates are involved in complex signaling cascades from plastids to the nucleus [22,23,24,25,26,27].

Once the biosynthetic steps of the MEP pathway were elucidated, researchers began to study the regulation of this pathway [28]. Regulatory control has been found to be exerted at different steps and different levels [24,29,30,31,32,33] and to vary under different environmental conditions such as drought and herbivore attack [34,35]. Major insights came from studies on the model plant *Arabidopsis thaliana*, where diverse experimental evidence demonstrates that the first step, DXS, is a critical regulation point [29,36]. Analysis of transgenic lines with altered DXS levels show increased or decreased levels of various isoprenoids end products including chlorophylls, carotenoids or tocopherols [37,38]. Similar results have been obtained in other plants including tomato [39,40], potato [41], and ginkgo [42].

Most of the research on the regulation of the MEP pathway was done in herbaceous plants, which in general do not emit the C_5_ gas isoprene, derived directly from DMADP. Isoprene is a volatile organic compound (VOC) that represents a major fraction of all hydrocarbons released to the atmosphere [43,44,45]. Isoprene has long been studied as a compound that protects plants during high temperature and oxidative stress [46,47,48,49,50], and it has been featured in studies on regulation due to the ease of making rapid, repeatable measurements of its rate of release from a single plant [51]. Isoprene has been implicated in resilience to high temperature by stabilizing chloroplast membranes [52] due to its suggested ability to partition in the lipid phase [53]. Isoprene has also been proposed to quench reactive oxygen species, such as hydrogen peroxide [54], singlet oxygen [55], and reactive nitrogen species [56]. For example, the inhibition of isoprene emission after fosmidomycin treatment of *Phragmites australis* increases hydrogen peroxide by up to 45% [57]. 

Isoprene emission from plants is well documented to be highly temperature, light, and CO_2_ dependent [53,54,58] and to be affected by stress conditions like drought or nutritional constraints [59,60,61]. In addition, it has been known for many years that in trees such as aspen and poplar, isoprene may represent the bulk of the MEP pathway carbon flux in mature leaves [62]. Thus, measuring the isoprene emission rate could be a powerful tool to analyze carbon flux through the MEP pathway. To follow the flux into isoprene, labeling methods have been employed, such as exposing plants to ^13^CO_2_, since in photosynthetically-active leaves, the MEP pathway uses metabolites formed from newly fixed carbon as a substrate [30,34,51,63,64]. Flux measurements in combination with determining the activities of specific enzymes in the MEP pathway can be used to calculate flux control coefficients (FCCs), which indicates the degree of control that each enzyme exerts on flux through a metabolic pathway [65,66,67,68]. Measurement of FCC typically involves the manipulation of enzymatic activity by genetic or biochemical methods to determine the effect of fractional changes in activity on fractional changes in carbon flux or metabolite concentration.

Here, we estimated the flux control of DXS in the MEP pathway under different environmental conditions in young leaves of grey poplar trees (*Populus* × *canescens*) via in vivo ^13^C-labeling. We established transgenic DXS lines and measured their isoprene emissions, the incorporation of isotopic label into isoprene, and the pool sizes of the main metabolites from the MEP pathway (DXP, MEcDP, and IDP+DMADP), which allowed us to calculate the carbon flux through the MEP pathway and then the FCC of DXS. In the DXS transgenic lines, we not only measured isoprene emission, but also the accumulation of non-volatile isoprenoids, including the chlorophylls and major carotenoids. Using these two approaches, we conclude that DXS is not a major rate-controlling enzyme of the MEP pathway in photosynthetically-active leaves of poplar. 

## 2. Results

### 2.1. Varying Light and Temperature Conditions Have Significant Effects on Isoprene Emission 

To determine if variable MEP pathway flux rates in grey poplar could be achieved under different environmental conditions, we first measured how light and temperature affect a well-known marker for MEP pathway flux—the isoprene emission. Light and temperature have frequently been described as influencing the emission of isoprene in other species [54,58,69]. In grey poplar, we measured isoprene emission under light intensities of 1000 or 250 µmol m^−2^ s^−1^ of incident photosynthetically-active quantum flux density (PPFD) and temperatures of 30 °C or 21 °C. Isoprene emission varied significantly under the different environmental conditions tested (*p* = 0.00307, one-way ANOVA), showing a higher emission rate under high light (1000 PPFD) and high temperature (30 °C) conditions, compared with low light (250 PPFD) and low temperature (21 °C) conditions (Figure 2).

### 2.2. The MEP Pathway to Isoprene Maintains Metabolic Steady State during ^13^CO_2_ Labeling

Next, we evaluated whether ^13^CO_2_ labeling in grey poplar leaves could be used to track the carbon flux in the MEP pathway. For each of the environmental conditions tested above, short-term kinetic labeling was performed in a dynamic flow cuvette with a proton transfer reaction mass spectrometer (PTR-MS) and a portable photosynthesis system attached. The incorporation of a ^13^C label into isoprene is depicted for one of these conditions in Figure 3. The ^13^C incorporation was evident after only a few minutes. The isoprene isotopologue with all five carbon atoms labeled became the major species present at about 10 min. Total ^13^C incorporation in the isoprene reached between 85–95% of labeling after 20 min in a ^13^CO_2_ atmosphere (Figure 3). 

Isoprene is formed in a single enzyme-catalyzed step from the last metabolite of the MEP pathway, DMADP. Our measurements of the ^13^C’s incorporation into the chromatographically-inseparable IDP+DMADP pool after 50 min under ^13^CO_2_ atmosphere show label incorporation similar to that of isoprene (p_13C_ incorporation = 0.941, two-way ANOVA) (Figure 4), making isoprene emission an accurate tool to track carbon flux in the MEP pathway of grey poplar leaves, as in other species [70,71]. In addition to being a product of the MEP pathway in the plastids, DMADP is also produced in the cytosol via the mevalonate pathway from acetyl-CoA. The fact that 85–95% of the total IDP+DMADP pool in the cell can be labeled via ^13^CO_2_ suggests that the mevalonate pathway is scarcely active in the leaves measured. An active mevalonate pathway would reduce the incorporation in IDP+DMADP of the label from ^13^CO_2_, but not affect ^13^C labeling in isoprene, which is produced in the plastids from DMADP derived from the MEP pathway [72,73,74].

### 2.3. DXS Overexpression or RNA Interference Have No Effect on Transcript Levels of Other MEP or Mevalonate Pathway Genes

We created transgenic grey poplar lines with altered DXS activity to check the contribution of this enzyme to MEP pathway flux and to look for changes in the accumulation of isoprenoid end products derived from the MEP pathway. Plant lines with silenced (iRNA-DXS1) and overexpressing (DXS1+) *PcDXS1* genes were compared to wild-type (WT) and empty vector (EV) lines by quantitative PCR. Since genes in the MEP pathway are known to follow a diurnal rhythm in plants [75,76], we measured midday transcript levels. In general, *PcDXS1* and *PcDXS2* (*p* < 0.001, one-way ANOVA), showed altered transcript levels among overexpression, silenced, and control lines. In *PcDXS1* silenced lines, both *PcDXS1* and *PcDXS2* transcript levels were 4-fold decreased compared to wild-type and empty vector control lines (*p* < 0.001, Tukey’s test) (Figure 5). Meanwhile, in the transgenic lines overexpressing *PcDXS1*, transcript levels of *PcDXS1* were 5–16 times increased (*p* < 0.001, Tukey’s test), while *PcDXS2* remained unaltered in comparison to the control plants (*p* > 0.05, Tukey’s test) (Figure 5). The expression of other selected MEP pathway genes (*PcDXR1 p* = 0.360, *PcDXR2 p* = 0.113, *PcCMK p* = 0.368; *PcHDR p* = 0.063, one-way ANOVAs), as well as mevalonate pathway genes (*PcHMGR p* = 0.209, *PcMVK p* = 0.147, one-way ANOVAs) was not affected by *PcDXS1* silencing or overexpression (Appendix A).

### 2.4. DXS-Silenced Lines Have Reduced DXS Enzyme Activity While Overexpression of DXS Increases Activity

To check whether *PcDXS* transcript changes cause changes in DXS enzyme activity, the condensation of pyruvate and glyceraldehyde 3-phosphate to form DXP was measured in vitro in extracts of leaves from the different genetic lines that had been kept under different conditions of light and temperature for 50 min. Statistical analysis showed that only the genetic background of the line affected the DXS activity (*p* < 0.001, two-way ANOVA), while the environmental conditions had no effect (*p* = 0.101, two-way ANOVA). After grouping all environmental conditions together, the DXS activity in silenced lines was 55% lower than the DXS activity in empty vector control plants (*p* < 0.001, Tukey’s test) (Figure 6), in correlation with the lower transcript levels of *PcDXS1* (Figure 5). On the other hand, overexpression lines showed an increase of 25% in DXS activity compared with the empty vector control plants (*p* = 0.0073, Tukey’s test) despite the much higher increase in *PcDXS* transcript levels measured by quantitative PCR (Figure 5).

### 2.5. The Rate of ^13^C Incorporation into Isoprene and the Pool Sizes of the Three Major Metabolites of the MEP Pathway Were Used to Calculate the Carbon Flux

We first checked that the transgenic DXS lines, like our wild-type line (Figure 2), varied their isoprene emission under different environmental conditions. Transgenic *DXS*-silenced (iRNA-DXS1) and overexpression lines (DXS1+) were compared to empty vector control plants by PTR-MS. Our results indeed showed a significant influence of the tested conditions (*p* < 0.0001, GLS model comparison), with an increased emission under higher light and temperature conditions. However, isoprene emission did not differ significantly between *DXS*-silenced or overexpression plants and the controls (*p* = 0.173, GLS model comparison, Figure 7), although DXS enzyme activity was 55% lower in silenced lines and increased by 25% in overexpression lines (Figure 5).

For flux calculations, it was also necessary to measure the pool sizes of DXP, MEcDP, and IDP+DMADP, the main metabolites in the MEP pathway. There were few differences between DXS transgenic and control lines, except under 250 PPFD and 30 °C, in which there was a significant increase in DXP pools in overexpressing lines compared to the controls (*p* = 0.0117, one-way ANOVA). There was also a significant increase in MEcDP pools in overexpressing lines compared to silenced lines, but not compared with empty vector controls (*p* = 0.0441, one-way ANOVAs with correction for multiple testing following Holm [78]) (Figure 8). On the other hand, under 1000 PPFD and 30 °C, the MEcDP pool was significantly increased in silenced and overexpression lines compared with empty vector controls (*p* = 0.0464 one-way ANOVA with correction for multiple testing following Holm [78]) (Figure 8). However, in the case of IDP+DMADP pools, we did not find any significant difference between transgenic lines under the various light and temperature conditions.

Information on the pool sizes of the MEP pathway intermediates was combined with data on ^13^CO_2_ incorporation into isoprene, as previously described, to calculate the carbon fluxes in the MEP pathway in the different genetic backgrounds (Table 1, Appendix A). Flux varied significantly under the different environmental conditions used, with higher carbon flux under higher light and temperature conditions (*p* < 0.001, two-way ANOVA) (Figure 9). There were also significant differences between control and transgenic plants (*p* < 0.001, two-way ANOVA), with a 25–35% increase in carbon flux in *DXS*-overexpressing lines compared to empty vector controls, and no differences between silenced lines and controls.

### 2.6. The Flux Control Coefficients for DXS Are Low under All Environmental Conditions Tested

The flux control coefficient (FCC) specifies the degree of control that each enzyme exerts on flux through a metabolic pathway. The calculated flux values of the transgenic *DXS*-silenced and overexpression lines and those of the empty vector control lines were plotted as a function of their corresponding enzyme activities in a double-logarithmic space (Appendix A) and the FCCs for the different environmental conditions were calculated from the slope of the linear regression of these data. In a linear pathway, the FCC value for a specific enzyme, which indicates the fractional change in flux due to a fractional change in that enzyme activity, has a range between 0 and 1. While an FCC value of 0 indicates that an alteration of enzyme activity will have no effect on flux, a value of 1 would correspond to a directly proportional relationship between enzyme concentration and pathway flux [79]. However, our experiments showed relatively low values for FCC of DXS under all environmental conditions tested (statistically indistinguishable from zero for all but the high light intensity-low temperature condition, where the FCC value was 0.18, see Appendix A). These results stand in marked contrast to the results previously reported for the FCC of DXS in Arabidopsis, which had an FCC of 0.82 [29].

### 2.7. Silencing DXS Results in Decreases in Some Isoprenoid End Products Compared with Overexpression Lines 

The role of DXS in controlling flux through the MEP pathway can also be estimated by measuring changes in levels of isoprenoid metabolites in plant lines with altered DXS activity. We analyzed the levels of several isoprenoids known to be produced from the MEP pathway, including the chlorophylls and major carotenoids (Figure 10). We found a significant 20–30% decrease in several compounds in PcDXS1-silenced lines compared with overexpression lines: lutein (*p* = 0.0213, one-way ANOVA; *p* = 0.0236 Tukey’s test), neoxanthin (*p* = 0.0103, one-way ANOVA; *p* = 0.0097, Tukey’s test), chlorophyll a (*p* = 0.0357, one-way ANOVA; *p* = 0.0325 Tukey’s test) and chlorophyll b (*p* = 0.0237, one-way ANOVA; *p* = 0.0236 Tukey’s test), but no significant differences between the other lines (*p* > 0.05, Tukey’s test). We did not find any overall differences in β-carotene (*p* = 0.412, one-way ANOVA) or violaxanthin (*p* = 0.0798, one-way ANOVA) (Figure 10).

## 3. Discussion

### 3.1. Photosynthetically Fixed Carbon Supplies the MEP Pathway in Grey Poplar Leaves

The localization of the MEP pathway in chloroplasts as well as other plastids suggests that photosynthetic products could be a major source of substrate for the pathway. Our ^13^CO_2_ labeling of photosynthetically-active grey poplar leaves supports the assertion that newly fixed carbon derived from CO_2_ through the Calvin-Benson cycle is the principal source of carbon for the MEP pathway, as measured by the emission of isoprene. The consecutive appearance of the labeled isotopologues of isoprene after only 5–20 min of ^13^CO_2_ in vivo feeding (Figure 3) demonstrates the close connection between photosynthesis and isoprene emission. The tight link between isoprene emission and the MEP pathway is confirmed by the very similar degree of labeling of the pathway metabolites IDP+DMADP and isoprene under all environmental conditions tested (Figure 4). The results also indicate that there is negligible exchange of IDP and DMADP between the plastid and the cytosol within the 50 min of total measurement after ^13^C labeling [29,51,73,74]. Previously reported ^13^CO_2_-feeding experiments showed that approximately 90% of the carbon atoms emitted as isoprene are rapidly labeled by ^13^C [80], although this percentage can vary with species [81] and leaf developmental stage [72]. It is also well known that isoprene emission is affected by light and temperature [69], and our data also show that changes in these conditions rapidly affect isoprene emission (Figure 7) as well as the metabolite pools of the MEP pathway (Figure 8). These results are completely consistent with previous work on the light dependency of the pathway [58,82]. 

Since the maximum amount of ^13^C label incorporation in isoprene was 90% (Figure 3, a small portion of the MEP pathway substrate might be supplied from carbon sources other than photosynthetic fixation [83]. In recent years, different sources have been proposed, such as starch breakdown in the chloroplast [84], deoxyxylulose phosphate derived from the pentose phosphate pathway [85], and extrachloroplastidic metabolites such as xylem-transported carbohydrates [86,87]. However, the lack of full 100% labeling of isoprene after ^13^CO_2_ administration could also arise from the lack of full labeling of the Calvin-Benson cycle by CO_2_ fixation due to a cytosolic shunt re-entering the cycle [88]. 

### 3.2. DXS Is Not a Major Rate-Controlling Enzyme of the MEP Pathway in Poplar 

To determine the influence of DXS on the MEP pathway in grey poplar, two approaches were followed. First, we created transgenic *DXS*-overexpressing and silenced lines and investigated whether these caused any alterations in MEP pathway flux compared to wild-type controls or controls transformed with only an empty vector. Overexpression of *DXS* resulted in nearly 10-fold increases of transcript levels, while silencing with interference RNA decreased transcript levels by approximately 4-fold (Figure 5). These changes in gene expression resulted in a 25% increase in DXS enzyme activity in overexpressing lines and a 50% decrease in activity in silenced lines (Figure 6). At the flux level, overexpressing lines displayed a 25–35% increase in flux compared to the empty vector controls, but were not significantly different from the silenced lines. Thus, DXS can be considered to have at most a modest effect on MEP pathway flux. 

Indeed, when we calculated flux control coefficients (FCCs), the fraction change in flux due to a fractional change in DXS activity, under four different light and temperature regimes, we obtained very low values. These values were not significantly different from zero for three regimes and only 0.18 for the fourth regime. FCC values range from 0 to 1, with 0 indicating that an alteration of enzyme activity has no effect on flux and a value of 1 indicating a direct proportional relationship between enzyme activity and pathway flux. In contrast to grey poplar, the calculated FCC for DXS in Arabidopsis was found to be 0.83 [29], suggesting that the MEP pathway in Arabidopsis is regulated in a different manner than that of grey poplar. Although FCCs for DXS have not been calculated for other plant species, a recent investigation of another woody plant, the gymnosperm *Picea glauca*, compared DXS activity and MEP pathway flux under different water regimes [34]. The close correlation of DXS activity and flux under well-watered and moderate drought conditions, but not under severe drought, indicates that the regulatory role of DXS may be contingent on environmental conditions.

As a second approach to studying the impact of DXS on MEP pathway flux, we analyzed the abundance of particular isoprenoid end products in our transgenic lines, such as isoprene, chlorophyll, and carotenoids, which are known to be produced out of building blocks supplied by the MEP pathway. Our measurements of isoprene demonstrated that DXS manipulation had no significant effect on emission. Although isoprene emission varied significantly under the different light and temperature conditions tested, no effect of the *DXS* genetic background on the isoprene was recorded. However, the results with chlorophylls and carotenoids showed a different trend. For both chlorophylls and two of the four carotenoids measured, there were 20–30% declines in *DXS*-silenced lines compared with *DXS* overexpression lines, but no difference between either silenced or overexpressing lines compared to the empty vector control. These results are consistent with previous reports showing changes in isoprenoid end products when *DXS* transcript levels are altered. For example, in Arabidopsis, overexpression increased and silencing decreased carotenoids, chlorophylls, α-tocopherol and the hormones GA and ABA, although the degree of increase was variable [37,51]. Overexpression also increased carotenoids in tomato fruit [39,40] and carotenoids and cytokinins in potato tubers [41]. In lavender, *DXS* overexpression increased the amount of monoterpene-dominated essential oil by over 3-fold, but the levels of chlorophylls and carotenoids were unchanged or decreased [38]. 

### 3.3. The MEP Pathway and the Role of Isoprene

Our results may also have some relevance for understanding the biological role of isoprene, which has remained enigmatic after many years of study. Researchers must explain why isoprene is a major sink for fixed carbon, for example up to 10% in mature poplar leaves, but is only released by approximately 20% of the world’s plant species [89,90]. While isoprene was long thought to quench reactive oxygen species and stabilize chloroplast membranes [91], calculations of the amounts of isoprene within the leaf revealed that this substance is present at concentrations too low to fulfill a protective role [92]. 

A more recent concept is that isoprene instead acts as a stress signal that alters gene expression [27], protein abundance [55,93], and hormone levels [56]. For example, the suppression of isoprene emission in gray poplar mediated by RNA interference (RNAi) resulted in the down-regulation of genes encoding enzymes and regulatory proteins involved in phenylpropanoid biosynthesis [52]. Similarly, transgenic tobacco emitting isoprene exhibited higher levels of phenylpropanoid accumulation compared to non-emitting wild-type controls [94]. Microarray analysis of Arabidopsis plants fumigated with a physiologically relevant concentration of isoprene revealed alterations in the expression of genes associated with various pathways, including those involved in stress response, such as phenylpropanoid biosynthesis [95]. In addition, Arabidopsis and tobacco engineered to emit isoprene demonstrated increased expression of genes related to stress tolerance [27].

From a different perspective, isoprene has been suggested to have a metabolic role, serving as a way for plants to decrease high amounts of DMADP while recovering the pyrophosphate moiety [96]. DMADP accumulation not only ties up large quantities of intracellular phosphate, but can also decrease MEP flux by feedback inhibition of DXS [33]. In fact, when isoprene formation is blocked at DMADP by silencing isoprene synthase, the flux rate through the MEP pathway is dramatically decreased [51]. The close correlation between DMADP and isoprene emission we found here is consistent with such a metabolic role for isoprene. 

## 4. Materials and Methods

### 4.1. Plant Material and Experimental Set-Up

All experiments were carried out with individual transgenic *DXS* grey poplar (*Populus* × *canescens*) and wild-type plants transformed with an empty vector as controls. Plants were amplified by micropropagation as described by [97]. Saplings of 10-cm height were repotted to soil (Klasmann potting substrate, Klasmann-Deilmann GmbH, Geeste, Germany) and propagated in a controlled environment chamber (day, 22 °C; night, 18 °C; 65% relative humidity; 16-h/8-h light/dark cycle) before they were transferred to the greenhouse (day, 23 to 25 °C; night, 19 to 23 °C; 50 to 60% relative humidity; 16-h/8-h light/dark cycle). Plants were then grown under these conditions until they reached a height of about 1.5 m (Appendix A).

A custom-built cuvette for a single *Populus* × *canescens* leaf was used together with a LI-6400XT Portable Photosynthesis System (Li-Cor Biosciences, Bad Homburg vor der Höhe, Germany) and a proton transfer reaction mass spectrometer (PTR-MS; Ionicon Analytik GmbH, Innsbruck, Austria) to monitor and record gas exchange parameters and isoprene emission in real time, and to carry out ^13^CO_2_ in vivo labeling. Ambient air was conditioned by passing through a wash bottle and a CO_2_ scrubber kept at 4 °C, providing a relative humidity of between 50–60% to the system. CO_2_ was maintained at a steady concentration with a supplemental CO_2_ gas cartridge. The cuvette held a single poplar leaf (13 cm diameter × 3 cm high) and delivered mixed air evenly across the leaf surface, helped by an integrated mixing fan and a Peltier element that maintained temperature. Previous studies have shown the importance of leaf stage development in isoprene emission [98,99], for this reason we used the 7th or 8th leaf in all plants to perform the experiments. Leaf temperature was monitored with a thermocouple in contact with the abaxial leaf surface. The total photosynthetic surface area was estimated by photography of a leaf together with a standardized area plot, both measured with Photoshop software (Version 12.0 ×64). Light was provided by an LED lamp localized above the cuvette, giving the required experimental light intensity monitored with a LI-250 hand-held quantum sensor (Li-Cor Biosciences). We conducted steady-state experiments under different light (PPFD = 1000 or 250 µmol m^−2^ s^−1^) and temperature (30 °C or 21 °C) conditions and 380 µmol mol^−1^ CO_2_ at a flow rate of 0.8 L min^−1^. Before starting the experiment with ^13^CO_2_, leaves were always acclimatized for at least 15 min in the cuvette until the release of water and CO_2_, isoprene emission and photosynthesis had reached steady-state conditions. In vivo labeling was performed for 50 min by a single change step to a labeling atmosphere that was identical to the acclimation conditions except that all CO_2_ was replaced with ^13^CO_2_ (380 µmol mol^−1^ > 99 atom % ^13^C, Linde, Munich, Germany). In similar experiments, we used leaves from intact plants to investigate the effect of DXS activity in the transgenic lines under different environmental conditions (PPFD = 1000 and 250 µmol m^−2^ s^−1^, temperature = 30 °C and 21 °C) on the MEP pathway. The outlet air was directed through the PTR-MS to determine the isotopic composition of the isoprene emission in real time. At the end of the labeling period, the leaf was quickly cut and flash frozen in liquid nitrogen. Each labeled leaf was ground to a fine powder in liquid nitrogen, keeping part of the tissue for RNA extraction and carotenoid analysis and lyophilizing the rest to dryness prior to analysis of label incorporation and characterization of DXS activity. Labeling experiments were restricted to a time period between 9:00 h and 16:00 h (summer time) in order to eliminate any influence of diurnal rhythm, which has been described for the MEP pathway [76,100].

### 4.2. Vector Construction and Transformation of Populus × canescens

In order to assess the effect of DXS on the MEP pathway, transgenic plants were made carrying an RNA interference (RNAi) construct specific to the two encoding genes, *PcDXS1* and *PcDXS2*. For that, a 305-bp region of the coding sequence was PCR-amplified by using the oligonucleotides PcDXS-RNAi-for (tgctctagagcatcatgctgcaatgggaggag) and PcDXS-RNAi-rev(cgggatcccggggggaggcatgccatgtaagt), then cloned in sense and antisense orientations into the multiple cloning sites of the pTRAIN vector on either side of an intron as described by [101]. After HindIII digestion, the excised RNAi cassette, including an upstream maize ubiquitin promotor, was ligated into the multiple cloning site of the pCAMBIA 1305.2 vector www.cambia.org (accessed on 11 June 2014). *Agrobacterium tumefaciens*-mediated stable transformation of the *P.* × *canescens* clone INRA 7171-B4 was performed following a protocol published by [102]. Transgenic RNAi poplar plants were amplified by micropropagation as described by [97]. To test the level of transgenicity, plants of four independent kanamycin-resistant transgenic lines plus an empty vector control line were characterized by qPCR, using four plants per line. For this, primer PcDXS-qPCR-for atgggaggagggacaggc and PcDXS-qPCR-rev gcagcaaagtaacagcatgctg were used.

### 4.3. Quantification and Label Incorporation Measurements of DXP, MEcDP and IDP/DMADP

The extraction and quantification of MEP metabolites was performed as described previously [103]. Briefly, 5 mg lyophilized material was extracted twice with a 250 µL solution of 50% (*v*/*v*) acetonitrile containing 10 mM ammonium acetate, pH 9, by vortexing for 5 min and centrifuging in a microcentrifuge for 5 min. 200 µL from each extraction was pooled in a new tube and dried under a nitrogen stream at 40 °C. The residue was dissolved in 100 µL ammonium acetate, pH 9, then extracted with 100 µL chloroform, and the phases separated by centrifugation. A 50 µL quantity of the upper phase was transferred to a new tube, diluted with 50 µL of acetonitrile, and centrifuged to remove any precipitate. The supernatant was transferred to an HPLC vial.

MEP pathway metabolites and their ^13^C incorporation were analyzed on an Agilent 1200 Infinity HPLC system (Agilent Technologies, Santa Clara, CA, USA) connected to an API 5000 triple quadrupole mass spectrometer (AB Sciex, Framingham, MA, USA). For separation, a HILIC-XBridge BEH Amide XP column (2.5 µm, 150 × 2.1 mm; Waters, Milford, MA, USA) together with a guard column containing the same sorbent (2.5 µm, 10 × 2.1 mm) and an SSI^TM^ high-pressure precolumn filter (Sigma-Aldrich, Taufkirchen, Germany) was used. The solvents used were 20 mM ammonium bicarbonate, pH 10.5, as solvent A and 80% (*v*/*v*) acetonitrile with 20 mM ammonium bicarbonate, pH 10.5 as solvent B; liquid-chromatography-mass spectrometry-grade ammonium hydroxide was used for pH adjustments. Separation was achieved with a flow rate of 500 µL min^−1^ and a column temperature of 25 °C, with a 5 µL sample injected. The solvent gradient profile started with a linear gradient from 0 to 16% A over 5 min, followed by an isocratic separation for another 5 min, a wash step at 40% A for 5 min, and a return to 0% A for 15 min of further equilibration. The mass spectrometer was used in the negative ionization mode with the following instrument settings: ion spray voltage −4500 V, turbo gas temperature 700 °C, nebulizer gas 70 psi, heating gas 30 psi, curtain gas 30 psi, and collision gas 10 psi. DXP and its isotope distribution were monitored by the following precursor ion → product ion reactions: *m*/*z* 212.9 → 96.9, *m*/*z* 213.9 → 96.9, *m*/*z* 214.9 → 96.9, *m*/*z* 215.9 → 96.9, *m*/*z* 216.9 → 96.9, *m*/*z* 217.9 → 96.9 (collision energy [CE], −16 V; declustering potential [DP], −60 V; and cell exit potential [CXP], −15 V). MEcDP and its distribution were monitored by the following precursor ion → product ion reactions: *m*/*z* 277.0 → 78.9, *m*/*z* 278.0 → 78.9, *m*/*z* 279.0 → 78.9, *m*/*z* 280.0 → 78.9, *m*/*z* 281.0 → 78.9, *m*/*z* 282.0 → 78.9, *m*/*z* (CE, −38 V; DP, −50 V; CXP, −11 V). IDP/DMADP and their isotope distributions were monitored by the following precursor ion → product ion reactions: *m*/*z* 244.9 → 78.9, *m*/*z* 245.9 → 78.9, *m*/*z* 246.9 → 78.9, *m*/*z* 247.9 → 78.9, *m*/*z* 248.9 → 78.9, *m*/*z* 249.9 → 78.9 (CE, −24 V; DP, −45 V; CXP, −11 V). Analyst 1.6 software (Applied Biosystems, Waltham, MA, USA) was used for data acquisition and processing.

The DXP, MEcDP and IDP+DMADP contents in plant extracts were quantified using external standard curves and normalized to unlabeled standards added in the approximate amount occurring in plant extracts after correction for natural ^13^C abundance. Normalization to added unlabeled standards was accomplished by analyzing each plant sample twice, once with and once without the addition of internal standards (ISTD) dissolved in 10 µL of water. The ITSD were added directly to the first extraction. To determine the amounts of metabolites originating from the plant material, the relative amounts of masses containing 3, 4, and 5 ^13^C atoms in the sample with added ISTD were compared with the values obtained without any ISTD. This accounted for losses due to the matrix effect during extraction and ion suppression effects in the mass spectrometer.

### 4.4. Measurement of Isoprene Emission and Determination of Stable ^13^C Isotopes of Isoprene with PTR-MS

The PTR-MS (Model 500, Ionicon Analytik GmbH, Innsbruck, Austria) was employed to measure isoprene emission and determine in real time the kinetic dynamics of ^12^C replaced with ^13^C in the isoprene molecule. The drift tube pressure was 2.2–2.3 mbar and the E/N ratio (electric field/particle density) was 130 Td (1 Td = 1 Townsend = 10^−17^ cm^2^ V ^−1^ s ^−1^. Isoprene was monitored with the mass signal *m*/*z* 69. The raw PTR-MS count-rate signal intensity (cps) was normalized (ncps) to the primary ion signal (hydronium (H_3_O^+^) *m*/*z* 21), hydronium dimer (H_3_O^+^· (H_2_O) *m*/*z* 37), and hydronium trimer (H_3_O^+^· (H_2_O)^2^
*m*/*z* 55) and drift tube pressure. The average of the normalized signal during the steady-state period was used to calculate the isoprene emission, after subtracting the background (empty chamber without leaf). Afterward, isoprene emission was normalized to the leaf area. In vivo labeling was accomplished by replacing the normal air (380 µmol mol^−1 12^CO_2_ including 1.1% ^13^CO_2_) entering the cuvette with labeling atmosphere (380 µmol mol^−1 13^CO_2_, 99.9%). Before initiating the labeling, single leaves were maintained for at least 15 min under normal air to ensure that isoprene emission and photosynthesis were stable. Labeling was performed for 50 min. The appearance of protonated masses of isoprene was followed in the PTR-MS by monitoring m/z 70 (^13^C_1_ ^12^C_4_H_9_), *m*/*z* 71 (^13^C_2_ ^12^C_3_H_9_), *m*/*z* 72 (^13^C_3_ ^12^C_2_H_9_), *m*/*z* 73 (^13^C_4_ ^12^C_1_H_9_), and *m*/*z* 74 (^13^C_5_H_9_). The percentage of ^13^C labeling was calculated by summing all ^13^C atoms incorporated in the isoprene isotopes, and dividing this number by the overall sum of unlabeled and labeled carbon atoms of isoprene [86]. 

### 4.5. Determination of Flux by Label Incorporation through the MEP Pathway and Isoprene

Plastidial concentrations of DXP, MEcDP, and IDP+DMADP were estimated by assuming that IDP+DMADP only occurred in the chloroplast, and that only plastidial DXP and MEcDP pools would be labeled on the time-scale of the labeling experiment (50 min). Thus, the plastidial concentrations of DXP and MEcDP were estimated by calculating the ratio of their final ^13^C-label incorporation to that of IDP+DMADP.

As isoprene labeling could be followed on-line instantaneously with the PTR-MS without the need for individual sampling, these measurements were taken as the instantaneous labeling state of the IDP+DMADP pool. This assumption was justified, since isoprene is produced from DMADP in a single enzymatic step, and the volatile isoprene gas escapes from the leaf. Moreover, the assumption was verified experimentally; the final label incorporation after 50 min in isoprene and in the IDP+DMADP pools was identical (Figure 4).

Following an approach similar to [104], the differential equations for label incorporation were integrated to obtain an analytical expression for the fractional labeling of the IDP+DMADP pool, with time as a function of the pool sizes of DXP, MEcDP, and IDP+DMADP, as well as the flux through the pathway:(1)$$f\left(t\right)=m\left\{1-\frac{A^{2}}{\left(A-B\right)\left(A-C\right)}\exp\left(-\frac{J}{A}t\right)-\frac{B^{2}}{\left(B-A\right)\left(B-C\right)}\exp\left(-\frac{J}{B}t\right)-\frac{C^{2}}{\left(C-A\right)\left(C-B\right)}\exp\left(-\frac{J}{C}t\right)\right\}$$
where *f*(*t*) is the fractional labeling of isoprene (which equates to the fractional labeling of IDP+DMADP) as a function of time, *m* is the maximal fractional labeling at the end of the run, *A*, *B*, and *C* are the pool sizes of DXP, MEcDP, and IDP+DMADP, respectively, *J* is the pathway flux and *t* is time. Equation (1) assumes that the pools of the other MEP pathway intermediates (MEP, ME-CDP, MEP-CDP, and HMBDP) are too small to significantly delay the label incorporation into downstream metabolites; these intermediates were below the limit of detection in the HPLC-MS analysis.

To calculate the flux, Equation (1) was fitted to isoprene labeling time-courses obtained from the PTR-MS [*f*(*t*)] (cf. Figure 3), with experimentally determined pool sizes of DXP, MEcDP, and IDP+DMADP (cf. Figure 8 and Table 1) entered as parameters *A*, *B*, and *C*, respectively. The parameters *m* and *J* were obtained by minimizing the sum of the squares of the differences between model and data with the Levenberg-Marquardt algorithm, as implemented in the Python LMFIT module [105]. The calculated fluxes are summarized in Figure 9 and Table 1.

### 4.6. Determination of In Vitro PcDXS Activity 

The DXS enzyme assay was performed by a method previously described in [106]. Briefly, activities were measured under saturating substrate conditions in total protein extracts of 5 mg of lyophilized plant tissues, extracted gently with 1 mL of extraction buffer (50 mM Tris-HCl, pH 8, 10% (*v*/*v*) glycerol, 0.5% (*v*/*v*) Tween 20, 1% (*w*/*v*) polyvinylpolypyrrolidone (average molecular weight = 360,000), 100 µM thiamine pyrophosphate, 10 mM DTT, 1 mM ascorbate, 2 mM imidazole, 1 mM sodium fluoride, 1.15 mM sodium molybdate, and 1% protease inhibitor cocktail (Sigma-Aldrich)) at 4 °C during 15 min in a rotating wheel followed by a centrifugation at 20,000× *g* for 20 min. A 30 µL aliquot of the supernatant was mixed with 70 µL of enzyme reaction buffer (50 mM Tris-HCl, pH 8, 10 mM MgCl_2_, 10% (*v*/*v*) glycerol, 2.5 mM DTT, 1 mM thiamine pyrophosphate, 10 mM pyruvate, 10 mM glyceraldehyde 3-phosphate, 2 mM imidazole, 1 mM sodium fluoride, 1.15 mM sodium molybdate, and 1% protease inhibitor cocktail) for 2 h at 25 °C. The enzyme reaction was stopped by vigorously vortexing for 5 min with 100 µL of chloroform and centrifuging at 20,000× *g* for 5 min. 45 µL of the aqueous phase was transferred to an HPLC vial containing 5 µL of labeled [^13^C_5_]-DXP dissolved in water as the internal standard obtained as described in [103]). The analysis of DXP was carried out as described above. The DXP produced by the DXS enzyme reaction was quantified using external standard curves and normalized to the [^13^C_5_]-DXP internal standard.

### 4.7. Calculation of Flux Control Coefficients

Control coefficients were calculated as described previously in [67,107] from the general formula $C_{v}^{y}=\frac{dy/y}{dv/v}=\frac{d\log y}{d\log v}$ where $C_{v}^{y}$ is the flux control coefficient, y is the flux, and v is the DXS activity (determined as described above). Data from empty vector control lines and DXS transgenic lines were combined, and the term dlog⁡y/dlog⁡v was calculated from the linear regression of a plot of the flux against DXS activity in a double-logarithmic space.

### 4.8. RNA Extraction, cDNA Synthesis, and Quantitative Real-Time PCR

RNA was extracted from poplar leaves harvested after 50 min of labeling using the InviTrap Spin Plant RNA Mini Kit (Stratec Biomedical) following the protocols of the manufacturer, with an additional DNase treatment step (RNase-Free DNase Set; Qiagen, Hilden, Germany). The first washing step was realized with 300 µL of wash buffer R1, followed by a DNase treatment (30 Kunitz units in 80 µL volume; 10 µL of RNase free water, and 70 µL of buffer RDD) added to the column and incubated for 15 min at room temperature. The column was washed with an additional 300 µL of wash buffer R1 continuing with the manufacturer’s protocol. RNA samples were quantified by spectrophotometry (Thermo Scientific, Waltham, MA, USA, NanoDrop 2000). Complementary DNA (cDNA) was synthesized using 1 µg of RNA using Superscript II reverse transcriptase (Invitrogen, Thermo Scientific Waltham, MA, USA) and 50 pmol of oligo(dT)_12–18_ primer (Invitrogen) in a 20 µL reaction volume, and diluted 5-fold with sterile water. 

Primers for the gene encoding 1-deoxy-D-xylulose-5-phosphate synthase (*PcDXS1*, *PcDXS2*) were designed and tested for their specificity (for the primer sequences, see Appendix A). The primers for the genes 1-deoxy-D-xylulose-reductoisomerase (*PcDXR1*, *PcDXR2*), diphosphocytidylmethylerythritol reductase (*PcCMK*), and 4-hydroxy-3-methylbut-2-en-1-yl diphosphate reductase (*PcHDR*) from the MEP pathway, and 3-hydroxy-3-methylglutaryl coenzyme A reductase (*PcHMGR*) and MVA kinase (*PcMVK*) from the MVA pathway were as in [51]. Transcript abundance was measured in a 20 µL volume containing 10 µL of Brilliant III Ultra-Fast SYBR Green QPCR Master Mix (Agilent Technologies), 10 pmol of reverse and 10 pmol of forward primers, and approximately 100 ng of cDNA. The qRT-PCR was performed using wild-type samples as a calibrator in a CFX Connect Real-Time PCR Detection System (Bio-Rad, Hercules, CA, USA) using a three-step program, followed by melting curve analysis: preincubation at 95 °C for 5 min; amplification for 40 cycles (denaturation: 95 °C for 15 s, annealing: check Appendix A for each gene; extension: and 72 °C for 45 s); and melting analysis from 65 °C to 95 °C. Primer efficiency was determined by a standard curve of a serial dilution of cDNA template from wild-type plants and calculated using the equation E = 10^(−1/slope)^. For normalization, specific primers of *P.* × *canescens Actin2* were used. Fold-change calculations were performed using the efficiency corrected model [77]. Biological replicates were assayed in triplicate, and each biological sample analyzed from three technical replicates.

### 4.9. HPLC Analysis of Plant Pigments

Plants from each transgenic or wild-type plant line were used to measure pigment content. Samples were protected from light and heat during these steps as well as during extraction. A 100 mg portion of freshly frozen tissue aliquots was extracted in 1.5 mL of acetone by shaking for 6 h at 4 °C in the dark, and after 5 min of centrifugation at 2350× *g* at 4 °C, 800 µL of the extract was transferred into a new light protected tube and 200 µL of water was added. After spinning the samples for 1 min at 5000 rpm at 4 °C, they were transferred to brown glass vials for analysis on an Agilent 1100 Series HPLC with a UV/VIS diode array detector. The detector was set at 445 nm for the detection of carotenoids and at 650 nm for the chlorophylls. The pigments were separated on a Supelcosil column LC-18 (7.5 cm × 4.6 mm × 3 µmm; Sigma-Aldrich) using an acetone (solvent A)/1 mM NaHCO_3_ (in water, solvent B) gradient with a flow rate of 1.5 mL min^−1^. The initial mobile phase consisted of 65/35% (*v*/*v*) solvent A/solvent B. Then, solvent A was linearly increased to 90% within 12 min and to 100% over 8 min. 100% solvent A was kept for 2 min and then decreased to 65% again within 3 min. Quantification was done using external standard curves. Authentic standards of the chlorophylls and β-carotene (Santa Cruz Biotechnology, Heidelberg, Germany) were analyzed in a range from 0.1 to 0.00625 mg ml^−1^. Lutein, neoxanthin, and violaxanthin were assumed to have the same response factor as β-carotene.

### 4.10. Statistical Analysis

The influence of environmental conditions on wild-type isoprene emission was compared with one-way ANOVA after square root transformation of the data. The influence of environmental conditions and genotype on flux, DXS activity, and MEP pool sizes was analyzed with two-way ANOVAs, followed by Tukey post hoc tests in cases of significant differences. In order to achieve normality of the residuals and homogeneity of variances, data were log-transformed.

The influence of environmental conditions and genotype on isoprene emission was investigated using the generalized least squares method (gls from the nlme library [108]) to account for the variance heterogeneity of the residuals. The varIdent variance structure was used. Whether the different variance of environmental conditions, genotype, or the combination of both factors should be incorporated into the model was determined by comparing models with different variance structures with a likelihood ratio test and choosing the model with the smallest AIC. In order to achieve normality of residuals, data were square root transformed. The influence (*p*-values) of the explanatory variables was determined by sequential removal of explanatory variables starting from the full model, and comparison of the simpler with the more complex model with a likelihood ratio test [109]. Differences between factor levels were determined by factor level reduction [110] All data were analyzed with R version 3.5.1 [111].

## 5. Conclusions

Our study contributes to the knowledge of how the MEP pathway is regulated in grey poplar leaves. We found that compared to Arabidopsis plants, the DXS enzyme plays only a minor role in controlling metabolic flux. Other MEP pathway enzymes play a more significant role. To gain a deeper understanding of MEP pathway regulation, further research is needed across a broader spectrum of plant species and within a variety of organs and tissue types, especially non-photosynthetic ones. Additionally, measurements should be conducted under a wider array of environmental conditions and track a more comprehensive range of isoprenoid products.

## Figures and Tables

**Figure 1 ijms-25-04181-f001:**
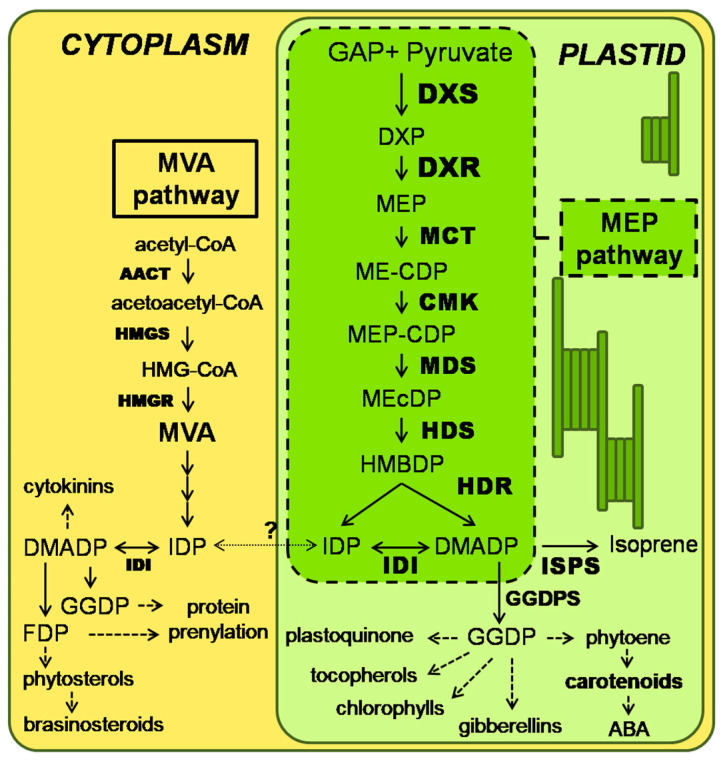
Scheme of the two isoprenoid biosynthesis pathways in the plant cell (modified after Rodriguez-Concepcion (2006) [14]). The MEP pathway is depicted in the bright green box inside the plastid. Dashed arrows indicate multiple steps. The question mark indicates that the extent of IDP+DMADP exchange between compartments is uncertain. MVA, mevalonic acid; AACT, acetoacetyl CoA thiolase; HMGS, hydroxymethylglutaryl (HMG) CoA synthase; HMGR, HMG-CoA reductase; IDP, isopentenyl diphosphate; DMADP, dimethylallyl diphosphate; IDI, IDP isomerase; GAP, glyceraldehyde phosphate; MEP, methylerythritol phosphate; DXP, deoxyxylulose phosphate; DXS, DXP synthase; DXR, DXP reductoisomerase; MCT, MEP cytidylyltransferase; ME-CDP, cytidine diphosphomethylerythritol; CMK, ME-CDP kinase; MEP-CDP, ME-CDP phosphate; MEcDP, methylerythritol cyclodiphosphate; MDS, MEcDP synthase; HMBDP, hydroxymethylbutenyl diphosphate; HDS, HMBDP synthase; HDR, HMBDP reductase; ISPS, isoprene synthase; GGDP, geranylgeranyl diphosphate; GGDPS, GGDP synthase; FDP, farnesyl diphosphate; ABA, abscisic acid.

**Figure 2 ijms-25-04181-f002:**
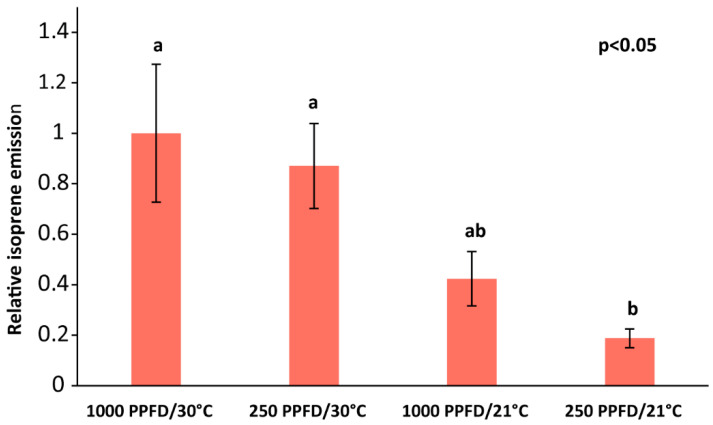
Effect of varying light and temperature conditions on isoprene emission in wild-type grey poplar under 1000 or 250 µmol m^−2^ s^−1^ of incident photosynthetically-active quantum flux density (PPFD) and a temperature of 30 °C or 21 °C and 380 µL L^−1^ CO_2_. Emission of isoprene was normalized to the leaf area. Means of *n* = 5 ± SE are shown. Significance differences (one-way ANOVA followed by Tukey’s test) at *p* < 0.05 are indicated with different letters.

**Figure 3 ijms-25-04181-f003:**
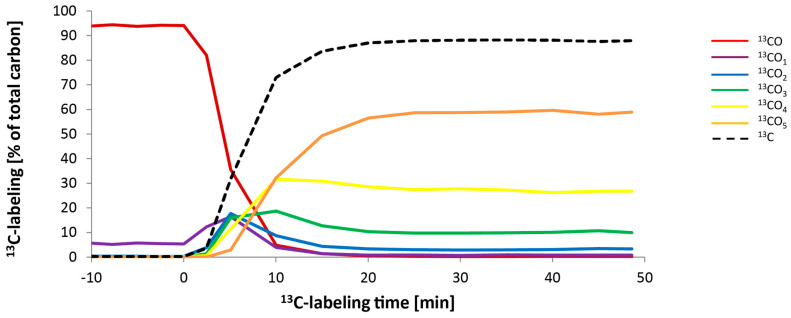
Dynamics of ^13^C incorporation into isoprene in WT *P.* × *canescens* leaves under one set of environmental conditions, 1000 PPFD and 30 °C. Leaves were fed for 50 min with 380 µL L^−1 13^CO_2_, 99.9%. ^13^C labeling began at 0 min. The isotopologue masses of isoprene are shown using different colors, representing the incorporation of different numbers of ^13^C-labeled carbon atoms: red, ^13^C_0_; purple, ^13^C_1_; blue, ^13^C_2_; green, ^13^C_3_; yellow, ^13^C_4_; orange, ^13^C_5_. The dashed black line represents the overall ^13^C labeling incorporation over the time.

**Figure 4 ijms-25-04181-f004:**
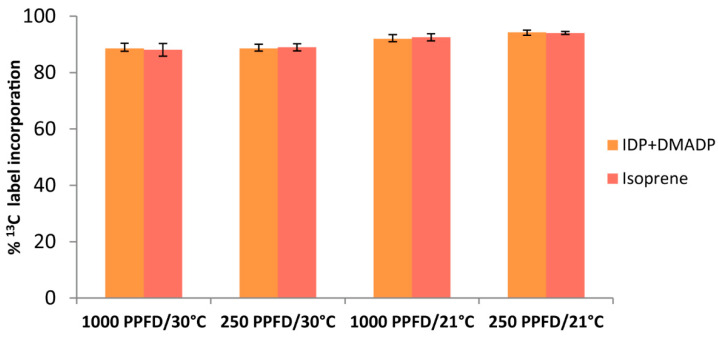
Incorporation of ^13^C into IDP+DMADP (orange) and isoprene (light red) in WT *P.* × *canescens* leaves after feeding them for 50 min with 380 µL L^−1 13^CO_2_, 99.9%, under different conditions of light (1000 or 250 PPFD) and temperature (30 or 21 °C). Shown are means (±SE) of five biological replicates.

**Figure 5 ijms-25-04181-f005:**
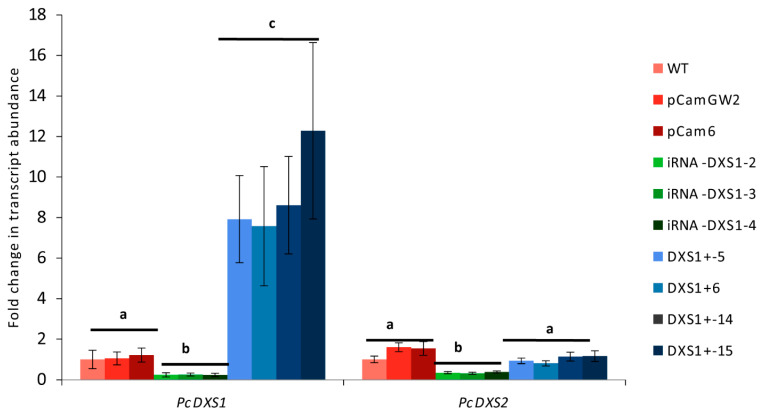
Transcript levels of grey poplar *PcDXS1* and *PcDXS2* genes at midday in transgenic silenced (iRNA-*DXS1-2*, iRNA-*DXS1-3*, iRNA-*DXS1-4*), and overexpression (*DXS1+-5*, *DXS1+-6*, *DXS1+-14*, *DXS1+-15*) lines compared with wild-type (WT) and empty vector (pCam) controls. Leaves were subjected to 250 µmol m^−2^ s^−1^ of incident PPFD, 21 °C leaf temperature and 380 µmol mol^−1^ of CO_2_ for 50 min before taking samples. Relative quantification was performed according to the efficiency corrected model [77]. Efficiencies were obtained from the slope of dilution curves using control cDNA diluted from 1 to 1024 times at 4× intervals. Target gene expression was normalized to *PcActin2* and fold-change values for each gene were calculated by comparison with the mean expression of the same gene in wild-type control plants (dark blue). Error bars indicate the SE of three biological replicates (*n* = 3) analyzed in triplicate SYBR green assays. Significance differences (one-way ANOVA followed by Tukey’s test) at *p* < 0.05 are indicated with different letters.

**Figure 6 ijms-25-04181-f006:**
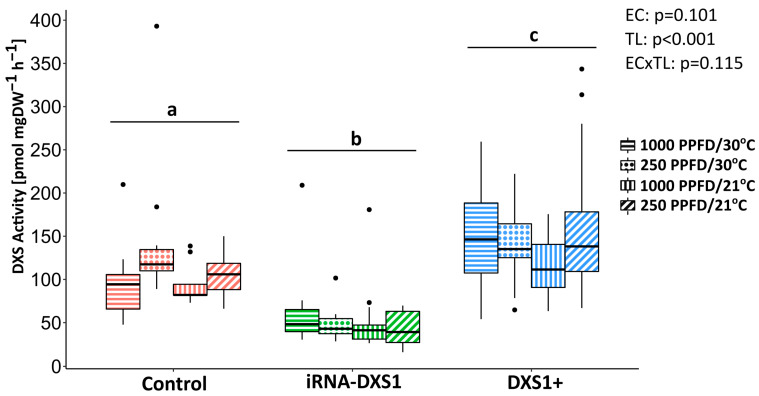
In vitro grey poplar DXS activity in *PcDXS1* transgenic silenced (green) and overexpression (blue) lines compared to empty vector control (red) lines. Activity was measured in vitro in protein extracts obtained from leaves grown under different conditions of light (1000 or 250 PPFD) and temperature (30 or 21 °C). The quantity of DXP produced was determined using an external standard curve and normalized to an internal standard of [^13^C_5_]DXP. Boxplots show medians, quartiles, and outliers. The sample size of each box is given in Table 1. Boxes are filled according the different environmental conditions tested. Significance differences (two-way ANOVA followed by Tukey’s test) at *p* < 0.05 are indicated with different letters. EC, environmental conditions; TL, transgenic lines.

**Figure 7 ijms-25-04181-f007:**
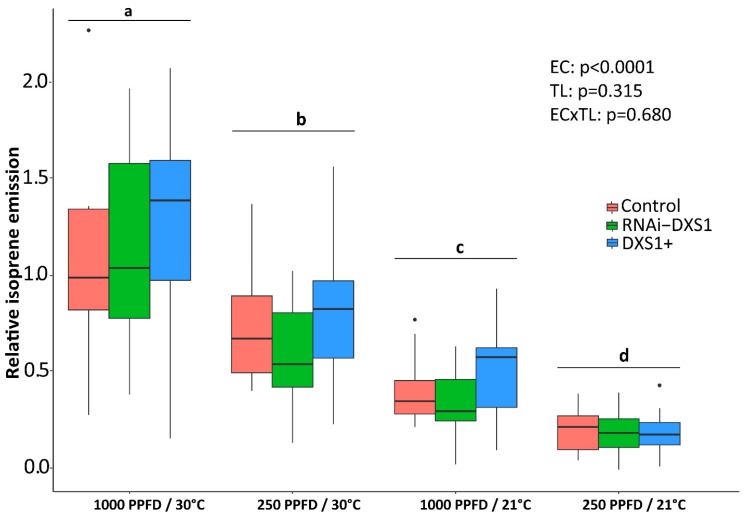
Isoprene emission of grey poplar from transgenic *DXS1*-silenced (iRNA-*DXS1*) and overexpression (*DXS1*+) lines compared to empty vector controls (Control), under different conditions of light (1000 or 250 PPFD) and temperature (30 or 21 °C). Boxplots show medians, quartiles, and outliers. T sample size of each box is given in Table 1. Significance differences (GLS model comparison after sequential factor removal) at *p* < 0.05 are indicated with different letters. EC, environmental conditions; TL, transgenic lines.

**Figure 8 ijms-25-04181-f008:**
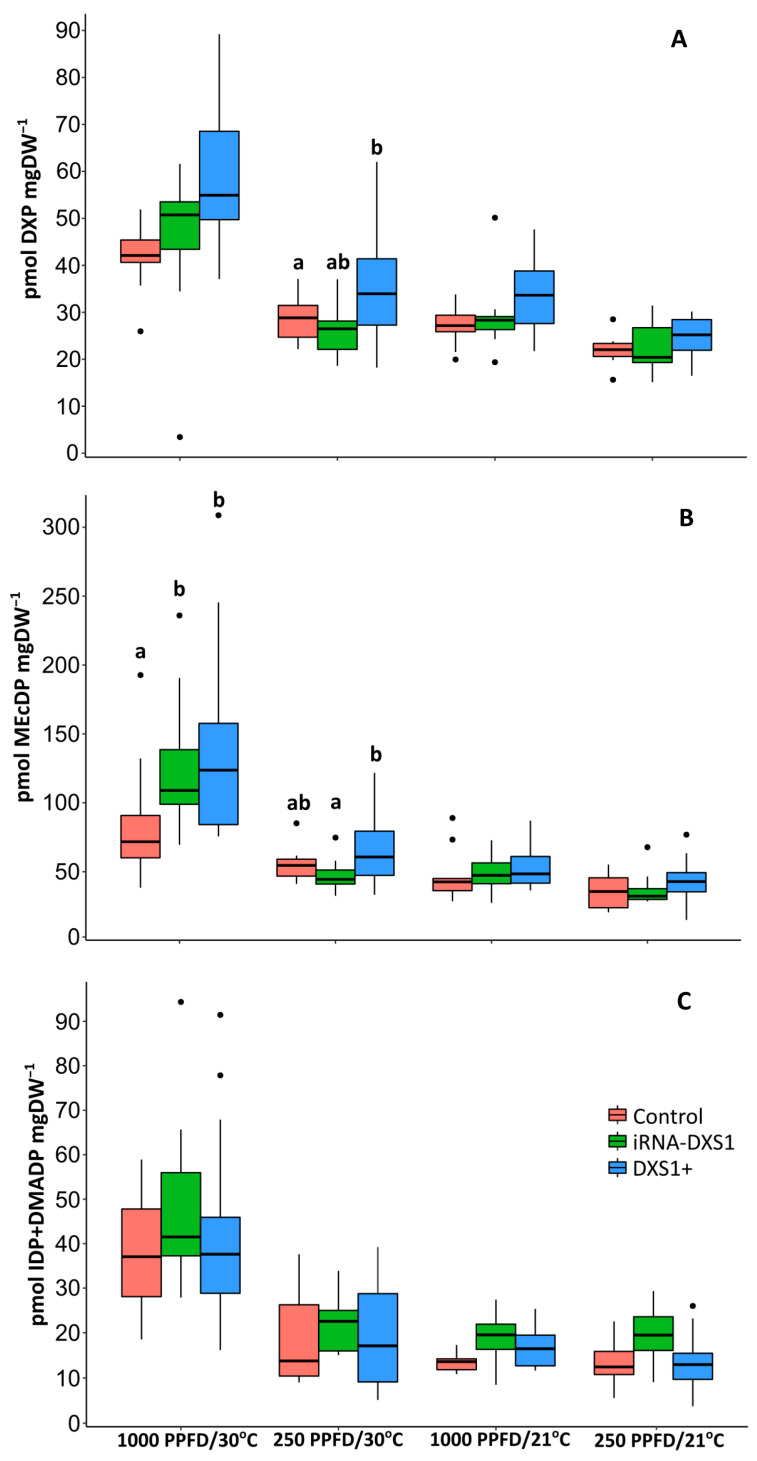
Pool sizes of (**A**) DXP, (**B**) MEcDP, and (**C**) IDP+DMADP in transgenic *DXS1*-silenced (RNAi-*DXS1*) and overexpression (*DXS1+*) lines compared to empty vector controls (Control) under different conditions of light (1000 or 250 PPFD) and temperature (30 or 21 °C). The concentrations of DXP, MEcDP and IDP+DMADP were determined using an external standard curve and normalized to internal unlabeled standards. Boxplots show medians, quartiles, and outliers. The sample size of each box is given in Table 1. Significance differences (one-way ANOVA followed by Tukey’s test with correction for multiple testing following Holm [78]) at *p* < 0.05 are indicated with different letters.

**Figure 9 ijms-25-04181-f009:**
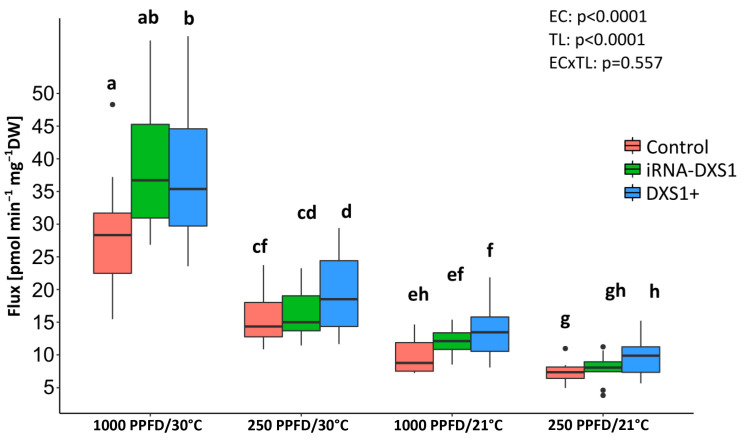
MEP pathway carbon flux in transgenic *DXS1*-silenced (iRNA-DXS1) and overexpression (*DXS1+*) lines under different conditions of light (1000 or 250 PPFD) and temperature (30 or 21 °C). Boxplots show medians, quartiles, and outliers. Sample size of each box is given in Table 1. Significant differences (two-way ANOVA followed by Tukey’s test) at *p* < 0.05 are indicated with different letters. EC, environmental conditions; TL, transgenic lines.

**Figure 10 ijms-25-04181-f010:**
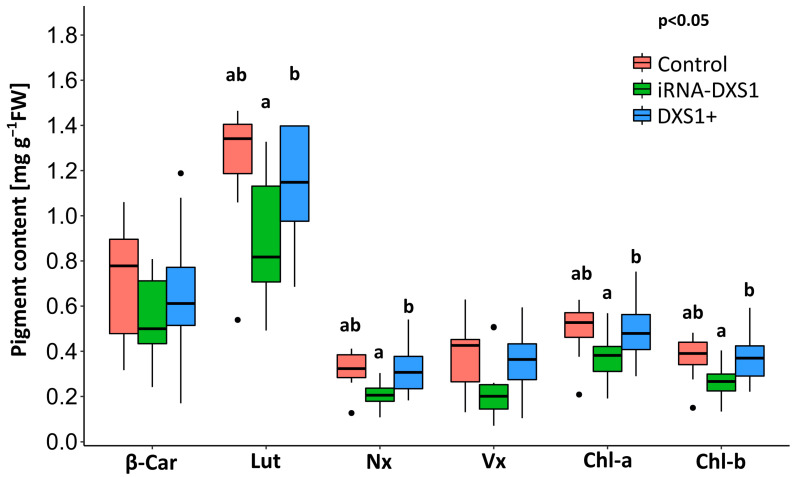
Content of carotenoids and chlorophylls in transgenic *DXS1*-silenced (iRNA-*DXS1*) and overexpression (*DXS1*+) lines. Leaves were acclimated under steady-state conditions (incident PPFD of 250 µmol m^−2^ s^−2^, leaf temperature 21 °C, and CO_2_ concentration of 380 µmol mol^−1^) before harvesting. Boxplots show medians, quartiles, and outliers. Sample sizes are given in Table 1. Significance differences between lines were tested with one-way ANOVA (Tukey’s test) at *p* < 0.05 and are indicated with different letters. ß-Car, ß-carotene; Lut, lutein; Nx, neoxanthin; Vx, violaxanthin; Chl-a, chlorophyll a; Chl-b, chlorophyll b.

**Table 1 ijms-25-04181-t001:** Comparison of carbon flux through the MEP pathway in poplar lines with different *DXS* expression determined under different environmental condition. Carbon flux through the MEP pathway was measured in transgenic *DXS1*-silenced (RNAi-*DXS1*) and overexpression (*DXS1+*) lines compared to empty vector controls (EV) under different conditions of light (1000 or 250 PPFD) and temperature (30 or 21 °C). The column “n” indicates the number of biological replicates. To calculate flux, the time course of ^13^C label incorporation into isoprene, which closely reflected the labeling of the IDP+DMADP pool (Figure 4), was fitted to a function that also took the pool sizes of DXP, MEcDP, and IDP+DMADP as input (see Section 4 for details).

Environmental Condition	Plant Line	n	Maximum Isoprene Labeling	DXP Pool Size	MEcDP Pool Size	IDP/DMADP Pool Size	Flux
				pmol mg^−1^ DW	pmol mg^−1^ DW	pmol mg^−1^ DW	pmol min^−1^ mg^−1^ DW
1000 PPFD 30 °C	EV	10	0.912 ± 0.007	41.65 ± 2.22	85.59 ± 14.43	38.09 ± 4.01	28.65 ± 2.95
	RNAi-*DXS1*	14	0.912 ± 0.005	46.01 ± 3.86	124.9 ± 12.14	47.74 ± 4.67	38.94 ± 2.62
	*DXS1*+	18	0.902 ± 0.006	58.61 ± 3.21	139.4 ± 15.89	41.20 ± 4.76	38.52 ± 2.56
250 PPFD 30 °C	EV	10	0.823 ± 0.036	28.53 ± 1.49	54.98 ± 4.06	18.47 ± 3.23	15.66 ± 1.42
	RNAi-*DXS1*	15	0.870 ± 0.008	25.61 ± 1.29	46.99 ± 2.66	22.03 ± 1.68	16.49 ± 0.96
	*DXS1*+	20	0.865 ± 0.018	35.97 ± 2.55	65.43 ± 5.23	19.41 ± 2.44	19.69 ± 1.24
1000 PPFD 21 °C	EV	9	0.932 ± 0.008	27.18 ± 1.45	48.57 ± 6.53	13.64 ± 0.78	9.79 ± 0.87
	RNAi-*DXS1*	13	0.929 ± 0.009	28.64 ± 1.96	48.19 ± 3.39	19.44 ± 1.35	12.15 ± 0.55
	*DXS1*+	16	0.943 ±0.005	33.51 ± 1.93	53.36 ± 3.79	16.90 ± 1.14	13.50 ± 0.93
250 PPFD 21 °C	EV	8	0.946 ± 0.009	21.99 ± 1.29	35.74 ± 4.54	13.31 ± 1.89	7.43 ± 0.65
	RNAi-*DXS1*	10	0.940 ± 0.005	22.29 ± 1.70	36.92 ± 3.85	19.57 ± 2.04	7.91 ± 0.73
	*DXS1*+	19	0.957 ± 0.006	24.59 ± 0.97	43.21 ± 3.37	13.18 ± 1.32	9.55 ± 0.64

## Data Availability

All data are contained within the article and Appendix A.

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
