# Peer review of "Deoxyxylulose 5-Phosphate Synthase Does Not Play a Major Role in Regulating the Methylerythritol 4-Phosphate Pathway in Poplar"

_ijms, 2024, doi:10.3390/ijms25084181_

Round 1

Reviewer 1 Report

Comments and Suggestions for Authors

The authors contributed to the study of the regulatory mechanism of MEP pathways. The authors examined the first enzyme in the plastid MEP pathway and rightly believe that studies of more plant species and their organs and tissues, as well as different environmental conditions, are needed to better understand the regulation of the MEP pathways. Some comments.

It is necessary to describe in more detail the biological role of isoprene, as well as its protective functions during osmotic stress and signaling.

 The title of Table 1 is missing.

 Provide a photo of the object (plant).

 It is necessary to give quantitative characteristics of ROS during the action of the protective functions of isoprene.

 In the Materials and Methods section, you need to edit subsection 4.7.

The Conclusion section is given separately.

 The list of references must be brought into compliance with the requirements of the journal.

Reviewer 2 Report

Comments and Suggestions for Authors

An interesting article however I have a few minor comments for the authors.

Firstly the title I feel is very specific and you have done much more than just measure DXP so I would re-vist the title and broaden it's appeal.

Please re-write this sentence in lines 103-105 it is not clear and involves some repetition:  Measurement of FCC typically involves manipulation of enzymatic activity by genetic or biochemical methods to determine the effect of fractional changes in activity on the fractional changes in flux or metabolite concentration.

Figure 2 Please include in the legend text what the isoprene emission is relative to. How has it been normalised?

Figure 3 please include a key (it is in the legend text but would be much clearer in the figure).

Figure 4 shows all 4 conditions (light/temp) however figure 3 shows only 1 please make sure the reasons for only showing 1 condition are clear either in the text or the legend.

Please standardise your figures. All figures show the conditions (light/temp) on the x axis apart from figure 6 which shows the lines on the x axis. This makes it a little confusing for the reader and it would be much clearer to change this figure to have conditions along the x axis.

I would move figure 10 to the supplementary. You can describe the findings in the text but as it shows no difference between types it does not need to be included in the main manuscript.
